# Effects of both Pro- and Synbiotics in Liver Surgery and Transplantation with Special Focus on the Gut–Liver Axis—A Systematic Review and Meta-Analysis

**DOI:** 10.3390/nu12082461

**Published:** 2020-08-15

**Authors:** Judith Kahn, Gudrun Pregartner, Peter Schemmer

**Affiliations:** 1General, Visceral, and Transplant Surgery, Department of Surgery, Medical University of Graz, 8036 Graz, Austria; judith.kahn@medunigraz.at; 2Institute for Medical Informatics, Statistics and Documentation, Medical University of Graz, 8036 Graz, Austria; gudrun.pregartner@medunigraz.at

**Keywords:** pro-/synbiotic, liver surgery, liver transplantation, gut–liver axis

## Abstract

The gut-liver axis is of upmost importance for the development of infections after surgery. Further bacterial translocation due to surgery-related dysbiosis is associated with limited detoxification function of the liver compromising outcome of surgical therapy. After liver surgery, about 30% of patients develop a bacterial infection, with the risk of bacteremia or even sepsis-associated liver failure and mortality in >40%. The potential benefit of pro-/synbiotics given before surgery is still under debate. Thus, a systematic literature search on trials comparing patients with or without supplementation and outcome after liver resection or transplantation was performed. Our search strategy revealed 12 relevant studies on perioperative administration of pro-/synbiotics in liver surgery. The pro-/synbiotic combinations and concentrations as well as administration timeframes differed between studies. Five studies were performed in liver transplantation and 7 in liver resection. All studies but one reported lower infection rates (pooled RR: 0.46, 95% CI: 0.31–0.67) with pro-/synbiotics. Liver function was assessed after LT/LR in 3 and 5 studies, respectively. Pro-/synbiotics improved function in 1/3 and 2/5 studies, respectively. Concluding, perioperative pro-/synbiotics clearly reduce infection after liver surgery. However, standard protocols with both well-defined probiotic strain preparations and administration timeframes are pending.

## 1. Introduction

Liver cirrhosis and hepatic tumors are often the final stage of a chronic liver disease. Surgical therapy, either liver resection (LR) or liver transplantation (LT) can be regarded as therapy of choice in most cases.

LR, which is especially performed for tumors, is associated with both mortality and morbidity of 3.5% and 10–15%, respectively [1,2,3,4,5,6,7,8,9,10,11,12,13,14,15]. About 30% of patients develop a bacterial infection and about 10% intra-abdominal sepsis, mostly caused by enterogenic bacteria after LR. The incidence of bacterial infections increases to up to 45% after extensive LR. With bacteremia, the risk of liver failure increases to >50% with a mortality of >40% [1,14,15]. Postoperative infections are the main reason for both morbidity and mortality, which are also associated with high treatment costs [16]. General risk factors include malnutrition and parenteral nutrition. The surgical trauma per se, direct manipulation of the intestine during abdominal surgery, reduced postoperative intestinal motility, and a limited detoxification function of a diseased liver decrease outcome quality of surgery. Even perioperative antibiotics as well as analgesics and proton pump inhibitors support a dysbiosis in the gut, which further increases the risk of infection.

Preoperative malnutrition especially increases infectious complications after LT [17], and decreased liver function before LT may be related to postoperative bacteremia [18]. Immunosuppression after LT per se further potentiates this risk for infections. Stress-related dysbiosis leads to bacterial translocation and, as a result, increased susceptibility to infection. The frequency of sepsis is increasing despite progressive antibiotic therapy and implementation of infection control policies [19,20,21,22]. A global concern about antimicrobial resistances emphasizes the necessity of new strategies to reduce the risk of infections in surgical patients.

Probiotics, prebiotics, and synbiotics, a combination of the previous two, are nutritional adjuncts and emerge as new therapeutic options for the prevention of surgical infections. Probiotics have been shown to be useful in the treatment of gastrointestinal infections, acute infectious diarrhea in children, traveler’s diarrhea, and antibiotic-associated diarrhea in both children and adults [23,24,25,26,27,28,29,30,31,32,33,34]. Mechanisms of action of probiotics include competitive exclusion of potentially pathogenic bacteria and direct antimicrobial effects, alteration of the pH of intestinal mucosa and prevention of bacterial translocation via tight junctions [35]. Furthermore, it has also been demonstrated that probiotics promote anti-inflammatory cytokine production [35]. Co-administration of prebiotics can enhance the proliferation of probiotic bacteria and certain bacterial genera can be stimulated selectively by these compounds.

There is a second important aspect with pro-/synbiotics in patients with chronic liver disease undergoing liver surgery: the crosstalk between the gut and its microbiota and the liver is increasingly recognized as so-called gut–liver axis. The gut and the liver communicate in a bidirectional way via the biliary tract, the portal venous system, and the systemic circulation, integrating signals generated by dietary, genetic, and environmental factors. The mucosa and vascular barrier of the intestine is the zone serving as a hub for the interactions between the gut and the liver, which is the central organ in host-metabolism. A dysbiosis with translocation of microbes and of microbial products when this barrier is disrupted may cause or worsen various hepatic diseases through this interdependence of gut and liver. Dysbiosis has been associated with many chronic liver conditions such as nonalcoholic fatty liver disease (NAFLD) or nonalcoholic steato-hepatitis (NASH), alcoholic liver disease (ALD), as well as cirrhosis and its complications like hepatic malignancy [36].

Current guidelines of the European Association for Study of the Liver (EASL) and the American Association for the Study of Liver Disease (AASLD) do not include pro-/synbiotics as part of the therapeutic protocol for LR and LT since their benefit is still under debate. Thus, a systematic review and meta-analysis was performed on the effects of pro-/synbiotic in liver surgery with a special focus on both infection and perioperative liver function.

## 2. Methods

### 2.1. Literature Search Strategy

A comprehensive systematic search of published articles on perioperative pro-/synbiotics in liver surgery from database inception to June 21, 2020, was performed using PubMed, CENTRAL and Embase (OvidSP). The search was carried out with the assistance of a librarian experienced in systematic reviews. A structured search strategy (Appendix A) was conducted with controlled vocabulary and relevant key terms to enhance sensitivity. The search strategy combined the following search terms: “probiotic OR probiot *”, “synbiotic OR synbiot *” AND “operation * OR surgical procedure* OR liver surgery OR liver surg * OR liver transplantation * OR hepatectomy OR liver resection OR liver resect *” AND “mortality OR morbidity OR sepsis OR surgical infection * OR surgery site infection * OR post-operative wound infection * OR postoperative wound infection * OR complication *. In addition, reference lists of included papers and previous reviews were reviewed to identify potentially eligible studies.

### 2.2. Study Selection

First, all abstracts identified by the search strategy after removal of duplicates were independently screened by two investigators (J.K., P.S.). If no abstract was available, the full text was obtained unless the article could be confidently excluded by title alone. Studies reporting on postoperative infection rate, which was the primary clinical efficacy outcome, or on parameters of liver function were considered. Randomized and non-randomized studies comparing perioperative administration of pro-/synbiotics in parallel groups were eligible. Case reports, case series, studies including children or animals, and in vitro studies were excluded.

Any disagreements during the screening process were resolved through discussion among the authors. We obtained the full texts of potentially eligible studies and again determined their suitability based on the selection criteria. Only full-text papers published in English were assessed.

### 2.3. Data Extraction

The following information was extracted from all studies: author, country, year of publication, study design, characteristics of the population studied, kind of liver surgery, number of study participants per group, time and duration of pro-/synbiotic administration, duration of follow-up, type of pro-/synbiotic used, infection rates and parameters of liver function.

### 2.4. Quality Assessment

The methodological quality of included randomized trials was evaluated with the Cochrane risk of bias assessment tool [37]. The methodological quality of the non-randomized studies included was assessed using the Newcastle–Ottawa Quality Assessment Scale for Cohort Studies [38].

### 2.5. Data Synthesis

We performed a random-effects meta-analysis with inverse variance weighting and the data are presented in forest plots. The risk ratio and the respective 95% confidence interval (CI) for therapy and control groups in each study were estimated from the reported events. We performed subgroup analyses by type of liver surgery (LT or LR) and a sensitivity analysis including only randomized trials.

The analysis was performed using R version 3.6.1, in particular package “meta”.

## 3. Results

### 3.1. Literature Search

The initial search identified 553 hits, 471 of which remained after the elimination of duplicates. A total of 426 publications were excluded during abstract screening. After the elimination of case reports (*n* = 2), animal studies (*n* = 3) and reviews (*n* = 28), 12 studies met the inclusion criteria [39,40,41,42,43,44,45,46,47,48,49,50] (Figure 1).

### 3.2. Study Characteristics

Twelve studies were included in the systematic review (Table 1). They were published between 2005 and 2017. A total of 9 different pro-/synbiotics were used in these studies. Study sites were in Japan (*n* = 5), Germany (*n* = 3), Poland, Australia, Bosnia, and China. The mean duration of pro-/synbiotic administration was 18.75 ± 9.75 (range 7–70) days. In 2 studies [39,45], probiotics were used, while synbiotics were administered in 10 trials [40,41,42,43,46,47,48,49]. The most commonly utilized comparators were placebo (*n* = 5) [39,40,41,45,47] and no intervention (*n* = 5) [42,44,46,48,50].

### 3.3. Study Quality

The study quality of the two non-randomized studies according to the Newcastle–Ottawa scale was 7 (out of a maximum of 9) and 8 points. The methodological quality of the 10 included randomized studies is presented in a risk of bias summary (Figure 2). As summarized there, the methodological quality was moderate; performance bias was detected in 10%, detection bias also in 10%, selection bias in 10–20%, attrition bias in 30% of the analyzed RCTs, whereas reporting bias was detected in 40%.

### 3.4. Infection Rate after Liver Transplantation

To date, a total of 5 studies, all of which were RCTs, have elucidated the mechanism of how intestinal microbiota influence the course after LT (Table 1). These studies primarily focused on infectious complications after LT. Four of these studies reported primarily post-transplant application of synbiotics [51]. One LT study reported pre-transplant synbiotic treatment starting at the time of wait-listing for LT and focused on the pre- and post-transplant patient outcomes including the development of the severity of liver disease pre-transplant, as assessed by the model of end-stage liver disease (MELD)- and Child Turcotte Pugh (CTP) scores and liver function post-transplant [39].

Rayes et al. compared the synbiotic *Lactobacillus plantarum 299* and inulin (group 1) with selective bowel decontamination (group 2) and inulin with heat-killed *Lactobacillus plantarum 299* (placebo, group 3) in 95 LT patients (31 patients in group 1, 32 patients in group 2, 32 patients in group 3) [40]. The lowest infection rates were shown with synbiotic treatment (13% vs. 48% (selective bowel decontamination), *p* = 0.017, and 34% (inulin alone)). Most infections were caused by enterogenic bacteria (Salmonella, Yersinia, Campylobacter). *Synbiotic 2000* was compared with a placebo containing only fibers in 66 (33/33) LT patients by Rayes et al. [41]. Significantly less bacterial infections occurred in the synbiotic group (3% vs. 48%). Patients in the synbiotic group were treated with antibiotics for a significantly shorter time (0.1 ± 0.1 days as compared to 3.8 ± 0.9 days). No severe adverse events occurred in both groups, and no infections were caused by prebiotics. Eguchi et al. compared synbiotics (*Bifidobacterium breve, Lactobacillus casei, Galactooligosaccharide*) with no therapy in 50 patients undergoing living donor liver transplantation (LDLT) [42] and reported significantly reduced infection rates. In the study of Zhang et al., synbiotics (*Bifidobacterium lactis, Lactobacillus acidophilus, casei, rhamnosus, brevis, lactis, plantarum* and fibers and enteral nutrition) were compared with enteral nutrition and fibers in 50 LT patients [43]. This study reported significantly lower bacterial infection rates within the synbiotic group (8.8% vs. 30.3%, *p* = 0.030). Grat et al. investigated the difference between probiotics and placebo, their administration starting at the time of wait-listing for LT in 55 patients (28 patients in probiotics group, 27 patients in placebo group) [39]. Treatment durations for <2 weeks, 2–10 weeks and >10 weeks were reported in 16.7% (4/24), 37.5% (9/24), and 45.8% (11/24) in the probiotic group and in 15.4% (4/26), 53.8% (14/26), and 30.8% (8/26) in the placebo group (*p* = 0.52). There was no difference in mortality rates but significantly lower infection rates in the probiotic group: 30-day-infection rate of 4.8% vs. 34.8% (*p* = 0.020), 90-day-infection rate of 4.8% vs. 47.8% (*p* = 0.002).

### 3.5. Infection Rate after Liver Resection

To date, only 7 studies have elucidated the mechanism of how intestinal microbiota influence the course after LR. Five of them were RCTs [44,45,47,48,49], and 2 were prospective non-randomized trials (Table 1) [46,50]. Most of the studies also focused primarily on postoperative infectious complications, which lead to increased morbidity and mortality, and also to high treatment costs. They further focused on intestinal integrity, microflora, and surgical outcomes [44,45], serum zonulin levels and intestinal integrity, systemic inflammatory response, microflora and surgical outcome, perioperative development of liver function and postoperative complications in one study [46], as well as liver regeneration after LR in one study [47]. Pro-/synbiotics were administered before and after LR in 5 of the studies, only post LR in 1 study [44], and only prior to surgery in 1 study [51]. The duration of pro-/synbiotic administration was between 1 and 14 days prior to LR, and 6 and 14 days following LR and for 1 month in the study with only preoperative administration [50]. Liu et al. compared probiotic vs. placebo in their double-blind RCT in 66 vs. 68 patients undergoing LR for colorectal metastases [45]. Lower infection rates were reported in the therapy group (*p* = 0.008), as well as lower endotoxin (*p* < 0.001) and zonulin levels (*p* = 0.004). Kanazawa et al. compared *Yakult BL Seichoyaku* and Galacto-oligosaccharides vs. no therapy in 44 (21 synbiotic group, 23 control group) patients undergoing LR [44]. All patients underwent hemihepatectomy or more extensive resection with en bloc resection of the caudate lobe and the extrahepatic bile duct. Combined vascular resection with reconstruction was performed in 7 patients in each of the two groups. Baseline characteristics were matched between the two groups [44]. Lower infection rates were reported in the synbiotics group (52.2% vs. 19%, *p* = 0.031). Usami et al. [48] also investigated infection rates in patients undergoing LR, either receiving synbiotic or no therapy, which were 0/32 in the synbiotic group and 5/29 (17.2%) in the control group. Rayes et al. compared prebiotic (beta-glucan, inulin, pectin, and resistant starch) with synbiotic (*P. pentosaceus, L. mesenteroides, L. plantarum, L. paracasei*, and beta-glucan, inulin, pectin, and resistant starch) in 19 patients (9 synbiotic group, 10 control group) [47]. Three of the patients in the synbiotic group and 2 of the control patients had an infectious complication.

Rifatbegovic et al. investigated patients with liver cirrhosis and histologically verified hepatocellular carcinoma (HCC) undergoing LR (segmentectomy/bisegmenctetomy, right and left hemihepatectomy/extended hemihepatectomy). They compared 60 patients with preoperative and postoperative use of synbiotics (*Lactobacillus plantarum, L. paracasei* subsp *paracasei, Pediacoccus pentoseceus, L. raffinolactis* and fibers) with 60 patients without synbiotic therapy [46]. Total protein and albumin were significantly higher in the synbiotic group. CRP, IL-1, IL-6, and TNF on post-operative days (POD) 7 and 14 were reported to be significantly lower in patients who used the synbiotic. The incidence of early and late post-operative complications after LR in the probiotic group was significantly smaller as compared with the control group (13.89 vs. 33.33%, *p* < 0.0001, and 8.33 vs. 19.44%, *p*< 0.001), and lower short-term and long-term mortality (8.33% vs. 2.78%, *p* < 0.02, 5.56% vs. 13.89%, *p* < 0.01) and significantly higher survival rates (86.11 vs. 66.67%) in the synbiotic group were reported in this study [46].

Sugawara et al. compared synbiotic use (*Lactobacillus casei* strain Shirota, *Bifidobacterium breve* strain Yakult as well as prebiotic galactooligosaccharides) 14 days preoperatively and 14 days postoperatively with only postoperative administration in 81 patients undergoing liver and extrahepatic bile duct resection with hepaticojejunostomy for perihilar cholangiocarcinoma [49]. Pre- and postoperative IL 6 levels were significantly lower, and the incidence of postoperative infectious complications was 30.0% vs. 12.1% (*p* = 0.049) in the group receiving the synbiotic pre- and postoperatively. Iida et al. investigated the influence of 1-month preoperative synbiotic therapy as compared with no intervention on infectious complications in 284 LR patients (*n* = 115 in synbiotic group, *n* = 169 in control group) [50]. They found no difference regarding infection rates after LR between the groups.

### 3.6. Meta-Analysis on Infection Rates

Risk ratios for infection rates of all patients receiving pro-/synbiotics perioperatively vs. controls were stratified by type of surgery (LT vs. LR) and displayed in a forest plot (Figure 3). Four studies (2 LT and 2 LR studies) found a significant difference between the groups [41,43,44,45], 1 LT study showed a trend favoring perioperative pro-/synbiotic use to reduce infection rates postoperatively [39].

Pooling the results for infection rates from studies of both study designs in a meta-analysis, pro-/synbiotics showed a significant lower infection rate (RR 0.46, 95% CI: 0.31–0.67). However, there was moderate heterogeneity (I^2^ = 43%, *p* = 0.06). When stratifying by LR or LT, both subgroups showed a benefit of pro-/synbiotics, but the effect was stronger in LT patients than in LR patients (Figure 3). Furthermore, residual heterogeneity was insubstantial. We also performed a sensitivity analysis including only RCTs, where one study was excluded. The results were very similar (Figure 4).

A meta-regression for mean age of study participants and duration of supplementation, adjusted for type of surgery, showed no significant influence of these covariates (*p* = 0.766 and *p* = 0.721, respectively).

### 3.7. Perioperative Liver Function Parameters in LT

Only 3 of the 5 LT studies reported perioperative liver function, with 1 study showing a significantly faster decrease of aspartate aminotransferase (AST) and alanine aminotransferase (ALT) after LT (Table 2) [39]. Rayes et al. reported no difference in bilirubin, AST, ALT, gamma glutamyl transferase (GGT), alkaline phosphatase (AP) after LT between the groups [40], no difference in laboratory markers of liver function, and also no difference in acute rejection rates [41]. Liver function was not reported in the studies by Eguchi et al. [42] and Zhang et al. [43]. Grat et al. [39] reported postoperative bilirubin levels to be significantly lower in the probiotics group (*p* = 0.020). The postoperative decrease of both AST and ALT activity was faster (*p* = 0.030), and international normalized ratio (INR) values were non-significantly lower in the probiotic group. There was one case of primary non-function (PNF) in the placebo group (4.3%) and no such case in the probiotic group (0.0%, n.s.) There was no waiting list mortality. In the placebo group, the MELD score significantly decreased from a median of 13 to 12.5 over the entire pre- transplant assessment period (*p* = 0.040). In the probiotic group, the MELD score changes were not significant. However, there was no significant difference in the MELD score change between the groups. No significant change in the CTP class over the entire period was noted in both the probiotic and placebo groups.

### 3.8. Perioperative Liver Function Parameters in LR

Liu et al. reported that both ALT and AST levels as measured on POD 10 were significantly lower with synbiotics. ALT levels (U/L) were 56.20 ± 18.16 in the control group vs. 36.28  ±  18.92 in the synbiotic group (*p* < 0.001); AST levels (U/L) were 45.62  ±  22.68 vs. 36.18  ±  21.52 (*p* < 0.015) (Table 2) [45]. Kanazawa et al. reported no difference with respect to postoperative serum total protein and serum total bilirubin levels between the 2 groups [44]. Usami et al. reported that total bilirubin, AST, and ALT were increased on POD 1–3 but returned to preoperative values by POD 13–15 in both groups, showing no difference between the groups (31 patients with synbiotics had a CTP score C in this study, and 28 patients in the control group, 1 patient each, had a CTP score B, and none CTP A) [48], but there was a trend toward higher levels of AST and ALT with synbiotics, which could partly be explained by longer operation times and more blood loss during surgery with synbiotics. Rayes et al. reported perioperative liver function capacity as measured by the ^13^C methacetin breath test and indocyanine green plasma disappearance rate, which was comparable in both groups. Complications had a negative impact on liver function, and complications were more severe in the synbiotic group in this study (Table 1; higher transaminases in the synbiotic group) [47]. Rifatbegovic et al. reported that AST, ALT, alkaline phosphatase (AP), and GGT preoperatively, on POD 7 and 14 with synbiotics for 10 days was significantly lower than in controls. Cholinesterase and total cholesterol did not differ between the groups; total and conjugated bilirubin was significantly lower with synbiotics. Perioperative liver function parameters were not reported in Sugawara’s study [38]. In Iida’s study, ALT, AST, total bilirubin, and prothrombin activity were measured before and after synbiotics-directly prior to surgery, after 30 days of synbiotics. Only patients with synbiotics (*n* = 115) were analyzed and have shown no difference in liver function after 1-month treatment with synbiotics as compared to the values before synbiotics (Table 2) [50]. Studies on perioperative pro-/synbiotics and their effect on changes of the gut microbiota are shown in Table 3.

## 4. Discussion

Surgical stress-related dysbiosis leads to bacterial translocation resulting in a limited detoxification function of the liver being the central hub of the gut–liver axis, increased susceptibility to infections, and worse outcome. After liver surgery, about 30% of the patients develop a bacterial infection, and if bacteremia occurs, the risk of liver failure increases to over 50% and mortality to over 40% [1,14,15].

Following surgery, systemic endotoxinemia is facilitated for the following reasons: the presence of aerobic and anaerobic microflora through the entire gastrointestinal tract, microvascular damage and diminished gut integrity resulting from ischemia/reperfusion injury, compromised and altered immunity, secondary to the inflammatory cascade, the duration of visceral ischemia (necrosis occurs much faster in intestinal mucosa than in other tissues), and the presence of hemorrhage and hypotension. Under these circumstances, the occurrence of transient endotoxinemia is almost certain, with infectious and non-infectious postoperative complications as possible consequences.

Unfortunately, postoperative infectious and non-infectious complications do occur in spite of preventive treatment with antibiotics immediately before or during surgical treatment and have so far been treated upon their occurrence during the postoperative recovery period.

Inflammation is normally a local and temporary event, which is restored upon its resolution. However, disrupted immune regulation can result in continuous pro-inflammatory cytokine activity and excessive or chronic inflammation. Sepsis is induced by an excessive response of the immune system to microorganisms or their products. LPS is a powerful activator of the innate immune system and interacts with macrophages, monocytes, neutrophils, and endothelial cells, which results in the rapid release of many pro- and anti-inflammatory mediators. LPS is the pivotal initiator of gram-negative sepsis, and the involvement of different receptors in this process has been demonstrated in various studies [52,53]. Intervening with this initiator by detoxification or neutralization therefore seems rational. It was shown that the enzyme alkaline phosphatase (AP) detoxifies LPS through dephosphorylation in vivo. In the liver, LPS can be inactivated and the initiation of inflammatory processes inhibited by activation of the enzyme AP by special bacterial strains (e.g., *Lactococcus lactis* W19).

The main functions of probiotics, pre-/synbiotics are the implementation of colonization resistance to pathogenic germs, improvement of bowel motility and splanchnic blood flow, the stimulation of enterocyte growth and mucus formation, modulation of intestinal inflammation, stabilization of the intestinal barrier, and stimulation of the immune system and also non-immune mechanisms through competition with potential pathogens [43]. A modulation of the intestinal functions of the immune system in the sense of reduction of pro-inflammatory cytokines, favoring tolerance-inducing cytokine profiles and regulatory pathways and an increase of the secretory IgA, is one of the multiple effects of probiotics. Epithelial cell homeostasis is promoted by increasing the barrier function, promoting cytoprotective responses, improving cell survival, and increasing mucin production. The effect of pathogenic germs is blocked by reducing the binding to the mucosa, lowering the pH in the lumen, and the production of antibacterial bacteriotoxins. It brings nutritional benefits by helping to break down otherwise indigestible food components and unlocking nutrients. Probiotics also have an impact on neuromodulation by expressing cannabinoid and µ-opioid receptors on the epithelial cells. This mechanism of action has been observed in *Lactobacillus acidophilus*, and these findings are based on murine models and evidence in humans is still lacking [54]. Apart from that, they reduce visceral hypersensitivity and the stress response. Of course, there is stress not only in the conventional sense but in connection with surgical interventions as a so-called “surgical stress reaction” [55].

A recent systematic review and meta-analysis on the effects of perioperative pro-/synbiotics in adults undergoing elective abdominal surgery by Chowdhury et al. showed that the perioperative administration of either probiotics or synbiotics significantly reduced the risk of postoperative infectious complications (relative risk (RR) 0.56; 95% confidence interval (CI) 0.46–0.69; *P* < 0.0001) [56]. Synbiotics showed greater effect on postoperative infections compared with probiotics alone (synbiotics RR: 0.46; 95% CI: 0.33–0.66; *P* < 0.0001, probiotics RR: 0.65; 95% CI: 0.53–0.80; *P* < 0.0001). Synbiotics but not probiotics also led to a reduction in total length of stay. There were no significant differences in mortality or non-infectious complications between groups [56]. Another review made similar observations that pro-/synbiotics counteract surgical site infections (SSIs) and surgically related complications (SRCs) via modulating the gut-immune response and via the production of short chain fatty acids [57]. A previous review by Moran et al. 2012 had shown no significant benefit of synbiotics in elective abdominal surgery, but patients undergoing hepatopancreatobiliary surgery or LT showed significantly reduced postoperative infectious complications [58].

Here, we focus on liver surgery. The stage of malnutrition as risk factor for postoperative complications and infections is common in patients with liver cirrhosis undergoing LR or LT [59]. A crosstalk between gut and liver is generally recognized [60], where intestinal dysbiosis and increased permeability of the intestine lead to microbial overgrowth and increased translocation of bacteria and fungi as well es their products like endotoxins from gram-negative bacteria and beta-glucans from fungi into the portal venous system. In the liver, they are recognized by Kupffer cells and stellate cells and are directly inactivated under normal conditions. Under pathological conditions when the intestinal barrier is impaired with an increased translocation of microbial products, an inflammatory process is initiated leading to liver damage impairing hepatocyte function and detoxification potential, which may lead to inflammatory liver diseases. Lipopolysaccharides (LPS) trigger toll-like receptor 4 (TLR4) expressed by Kupffer and hepatic stellate cells to activate transforming growth factor β signaling that leads to the development of hepatic fibrosis and eventually cirrhosis [36]. In end-stage liver disease, microbial products that cannot be cleared by the liver get into the systemic circulation, activating immune cells and leading to damage of distant organs. Endotoxinemia primes neutrophils with development of small amounts of reactive oxygen species increasing bacterial clearance. Chronic over-stimulation can lead to exhaustion and reduction of phagocytic capacity and oxidative burst and increased infection rates. Bacterial infections in general are common complications resulting from impaired immune responsiveness.

Synbiotics in cirrhotic patients were reported to significantly decrease ammonia and serum endotoxin levels, as well as prevention of cecal overgrowth with *Escherichia coli* and *Staphylococcus* sp. It was also reported to lead to reversal of minimal hepatic encephalopathy and improvement of liver function in half of the patients. Fermentable fibers alone were also effective in some patients according to a study by Liu et al. [61].

A prospective randomized placebo-controlled, double-blind study was conducted at the Medical University Clinic of Graz; here, non-surgical patients with liver cirrhosis taking synbiotics were shown to have better liver function and less susceptibility to infection [62]. Liver function significantly improved as measured by the most common liver function scores (CTP (Child Turcotte Pugh) score, model of end stage liver disease (MELD) score) with synbiotics as compared with the placebo group, although the function of the verum group at the beginning of the study was significantly worse. The infection rate with synbiotics was significantly lower than in controls. The tolerance and the compliance were excellent, and no interactions with other drugs were reported.

Only 1 of the LT studies reported a significant faster postoperative decrease of bilirubin levels with probiotics (*p* = 0.020), postoperative decrease of AST and ALT activities (*p* = 0.030), as well as faster improvement of liver synthesis parameters with lower INR values. They also reported the dynamics of the MELD and CTP scores of the patients on the waiting list, which showed no difference between the groups. This study by Grat et al. investigated preoperative probiotics from the time of wait-listing for LT on [39]. Two LT studies by Rayes et al. reported no difference regarding parameters of liver synthesis and liver function in LT patients; the administration time frames were between pre-transplant day 1 and post-transplant day 12, as well as post-transplant day 1 and 14. The 2 other LT studies did not report liver function.

The incidence of postoperative infections is high in cirrhotic patients undergoing LR for liver tumors. Bacterial translocation after LR in patients with liver cirrhosis has been reported to occur in mesenteric lymph nodes in 20% [63], 44% of those patients with positive lymph nodes developed infectious complications, and 88% of the cases with infectious complications indicated the same bacterial species as the lymph node cultures.

The risk of infection is even higher after LT because of immunosuppression. A recent study on SSIs after LT [64] clarifies the explosiveness of this topic with a percentage of 53% infections that are caused by multi-resistant germs with significantly higher mortality after LT. The origin of deep SSI was abscesses in 58%, a peritonitis in 28%, deep wound infections in 8% and a cholangitis in 6%. An increase of 24 hospital days per patient, 159.967 US$ extra costs and a 10% higher mortality was reported. In LT, especially, other factors like malnutrition, ischemia-reperfusion injury, and immunosuppressive therapy may lead to dysbiosis, disrupted intestinal barrier, alterations in the innate immunity response, and bacterial translocation. This can be associated with early infections, graft failure, and decreased survival [60]. Due to a dysbiosis with a decrease of beneficial bacteria and an increase of pathogenic species, the response to the host consists of higher endotoxin levels and increased bacterial translocation. Previous studies suggest that the intestinal microbiota regulate liver tumorigenesis or inflammatory reactions through altering the activity of pro-inflammatory microorganism-associated molecular patterns, bacterial metabolites, natural killer (NK) T cells-mediated bile acid metabolism, and prostaglandin (PG)E2- mediated suppression of antitumor immunity [61,63,64,65]. In the liver, lipopolysaccharides (LPS) can be inactivated by activation of the enzyme alkaline phosphatase by special bacterial strains (e.g., lactococcus lactis W19). Probiotics can reduce the occurrence rate of bacterial infections by stabilizing the gastrointestinal barrier and by banishing pathogens. When taking safety of pro-/synbiotic use in immunocompromised and immunosuppressed patients into account, based on available literature, it may be prudent to avoid products containing *Saccharomyces* sp., because these products were not used in the efficacy studies. The majority of the adverse event reporting reports products containing *Saccharomyces* sp. [65].

Limitations of this review were that 3 studies (2 LT studies and 1 LR study) were performed by 1 investigator [40,41], Eguchi et al. had a retrospective control group [42], and 2 of the LR studies were prospective non-randomized studies. The combinations and concentrations of the pro-/synbiotic preparations were different, though *Lactobacillus* sp. was the main component. The results of the meta-analysis have to be interpreted with caution because the duration of perioperative exposure differed between studies, as well as the control groups being inconsistent (placebo vs. no therapy vs. administration of only prebiotics).

## 5. Conclusions

Perioperative pro-/synbiotics have been investigated in various RCTs in liver surgery with different primary study endpoints. Most of the studies focused on postoperative infections that lead to increased morbidity and mortality but also on intestinal integrity, on the microflora, as well as on the development of perioperative liver function and surgical outcomes. Pro-/synbiotics in liver surgery clearly reduce infections. However, current evidence on the effects of prophylactic perioperative administration of pro-/synbiotics in liver surgery is conflicting due to pending standardization of pro-/synbiotic preparations, duration of perioperative exposure, the route of administration as well as standardized study controls (standard care or placebo), making comparison of the trials challenging.

## Figures and Tables

**Figure 1 nutrients-12-02461-f001:**
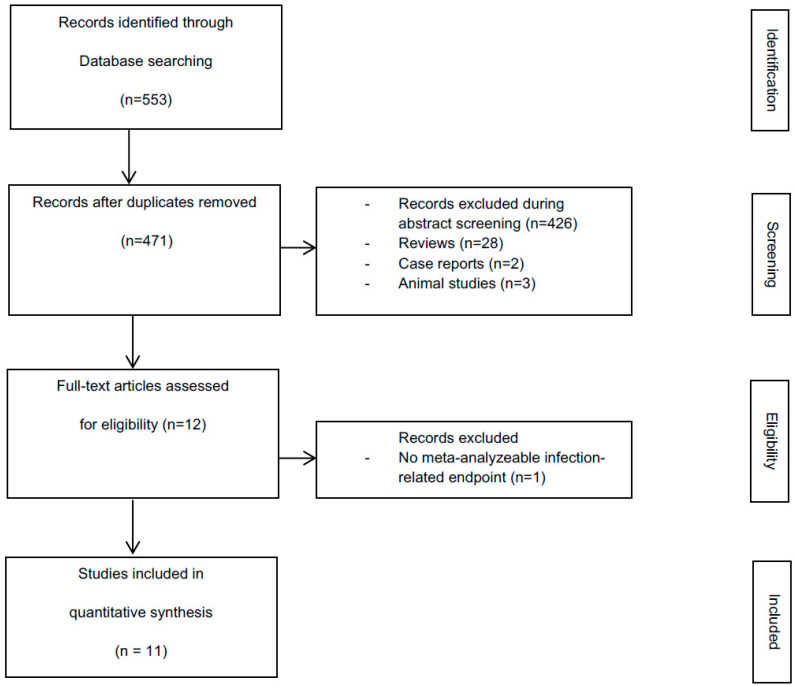
Flow chart depicting the screening and selection process for the systematic review of pro-/synbiotics on the prevention of infections and perioperative liver function.

**Figure 2 nutrients-12-02461-f002:**
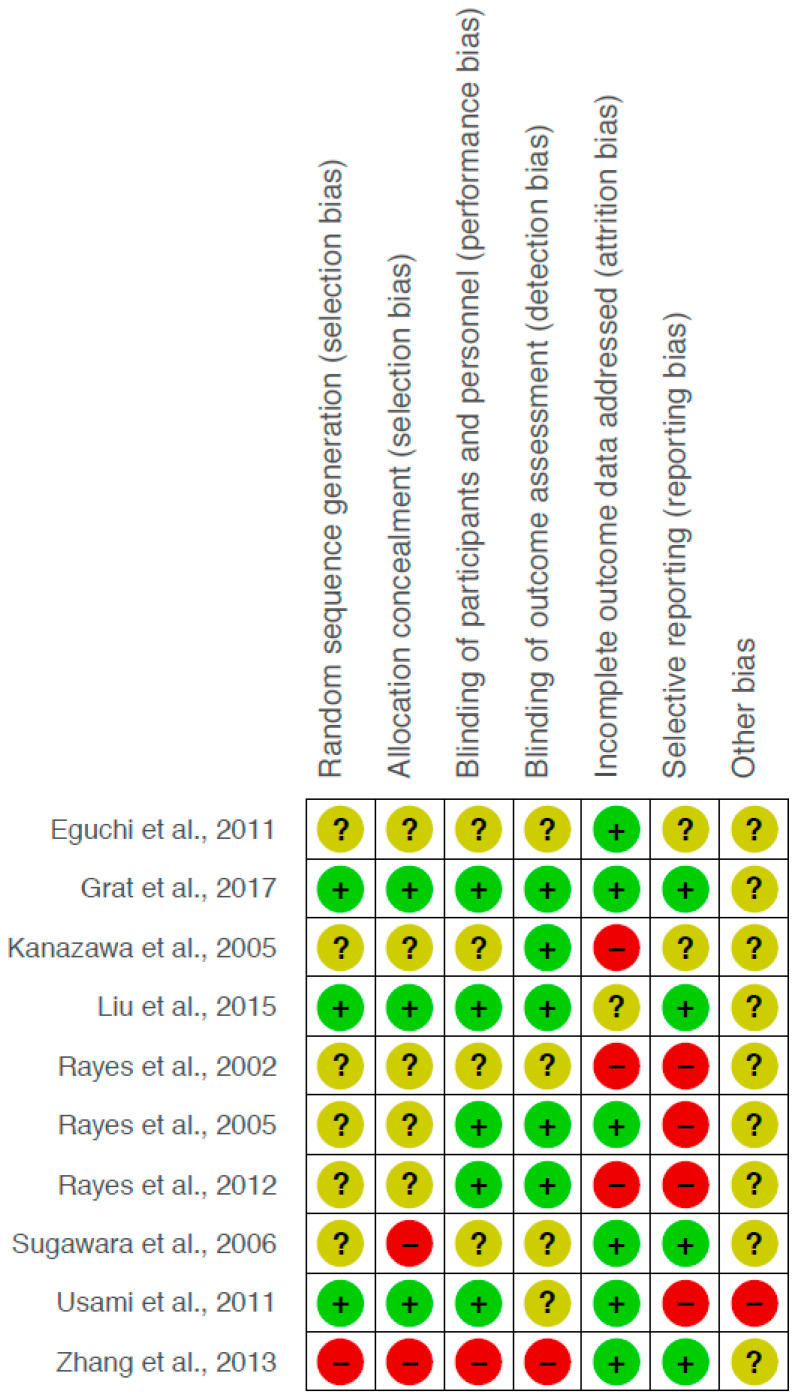
Risk of bias summary: review of authors’ judgement on the risk of bias for the analyzed randomized controlled trials. Red circle symbolizes high risk of bias, green circle symbolizes low risk of bias, yellow circle symbolizes unclear risk of bias.

**Figure 3 nutrients-12-02461-f003:**
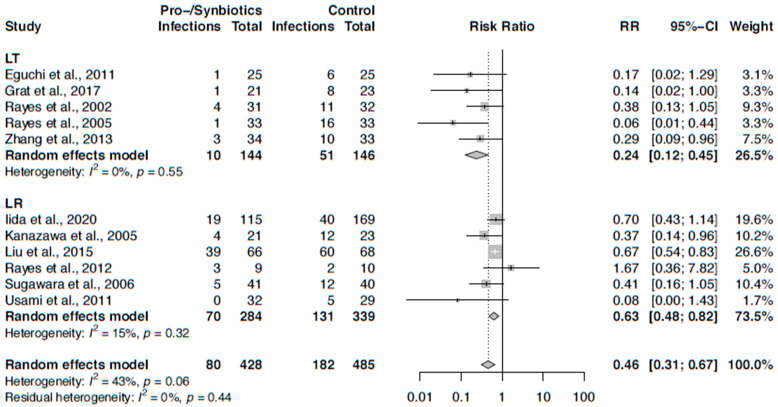
Forest plot of trials on perioperative pro-/synbiotics and their effect on infection stratified by type of surgery: liver resection (LR), liver transplantation (LT).

**Figure 4 nutrients-12-02461-f004:**
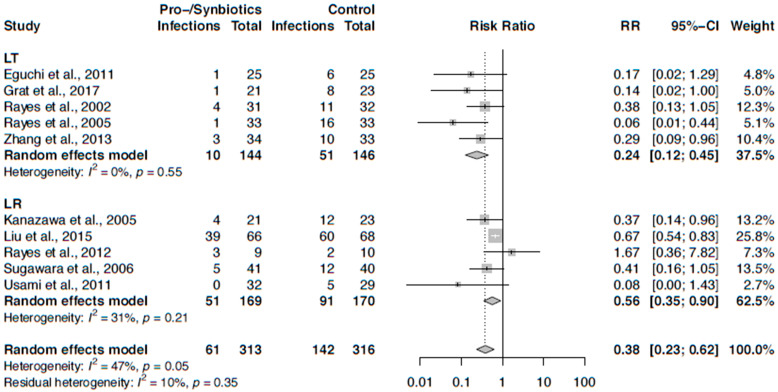
Forest plot of randomized controlled trials on perioperative pro-/synbiotics and their effect on infection stratified by type of surgery: liver resection (LR), liver transplantation (LT).

**Table 1 nutrients-12-02461-t001:** Studies on perioperative pro-/synbiotics and their effect on postoperative infection.

Study (Ref.)	Country	Study Population	Pro-/Synbiotic (*n*)/ Control (*n*)	Pro-/Synbiotics Used	Pro-/Synbiotic Content and Pharmaceutical Form	Control Used	Time of Administration	Results Pro-Synbiotic/Control Postoperative Infection Rate	Age (Years) All pt.; Pro-/Synbiotic/ Control	Study Design
**Liver transplantation (LT)**		PA/PE *n* = 290/343								
**Rayes et al., 2002 [40]**	Germany	63/105	31/32	Synbiotics (*Lactobacillus plantarum* 299 and inulin with selective bowel decontamination and enteric nutrition)	10^9^ CFU 2×/day	Placebo/inulin	Just before LT until 12 days post LT (13 days)	4/31, 11/32 FU POD 13	50 ± 2; 50 ± 2/ 50 ± 2	RCT
**Rayes et al., 2005 [41]**	Germany	66/66	33/33	Synbiotics (*Pediococcus pentosaceus*, *Leuconostoc mesenteroides*, *L. paracasei*, 1010 *L. plantarum* 2362, beta-glucan, inulin, pectin, and resistant starch and enteric nutrition)	probiotic: 10^10^ CFU, prebiotic: 10 g 2×/day; powder	Placebo/fibers	Just after LT until 14 days post LT (13 days)	1/33, 16/33 FU POD 30	51.5 ± 2; 53 ± 2/ 50 ± 2	RCT
**Eguchi et al., 2011 [42]**	Japan	50/50	25/25	Synbiotics (*Bifidobacte rium breve*, *Lactobacillus casei*, Galactooligo saccharide and enteric nutrition)	15 mg 20 mg 15 mg 3×/day	No intervention, enteric nutrition	2 days prior to LDLT until 14 days post LDLT (16 days)	1/25, 6/25 FU POD 19	56.5 ± NR; 56 (33–66)/ 57 (25–68)	RCT
**Zhang et al., 2013 [43]**	Australia	67/67	34/33	Synbiotics (*Lactobacillus acidophilus, plantarum, lactis, casei, rhamnosus, brevis, Bifidobacterium lactis* and fibers and enteric nutrition)	15.5 × 10^9^; 5.0 × 10^9^; 2.0 × 10^9^; 1.5 × 10^9^; 1.5 × 10^9^; 1.5 × 10^9^ CFU; capsules	Enteric nutrition and fibers	Immediately after LT for 7 days at minimum (7 days)	3/34, 10/33 FU POD 8	56.01 ± 10.98; 57 ± 10/ 55 ± 12	RCT
**Grat et al.,** **2017 [39]**	Poland	44/55	21/23	Probiotics (*Lactococcus lactis, Lactobacillus casei, Lactobacillus acidophilus* and *Bifidobacterium bifidum*)	50% 25% 12.5% 12.5% 3 × 10^9^ CFU; capsules	Placebo	Starting at the time of wait-listing for LT until LT Treatment duration for <2 weeks, 2–10 weeks, and >10 weeks in 16.7% (4/24), 37.5% (9/24), and 45.8% (11/24) in the probiotic group and 15.4% (4/26), 53.8% (14/26), and 30.8% (8/26) in the placebo group	30-day infection rate 0.09 (95%CI 0.01–0.83) 1/21, 8/23 90-day infection rate 0.06 (95%CI 0.01–0.48) 1/21, 11/23 FU POD 90	50.95 ± NR; 52 (47 - 58)/50 (35 - 61)	RCT
**Liver resection (LR)**		*n* = 743/849								
**Kanazawa et al., 2005 [44]**	Japan	44/54	21/23	*Bifidobacterium breve* strain Yakult, *Lactobacillus casei* strain Shirota; prebiotic: galactooligo saccharides	1 × 10^8^/g 1 × 10^8^/g 3 g/day prebiotic: 12 g/day	No intervention, enteric nutrition	Just after LT until 14 days post LT (13 days)	4/21, 12/23 FU POD 30	63.75 ± 9.64; 62.5 ± 9.9/64.9 ± 9.4	RCT
**Liu et al.,** **2015 [45]**	China	134/150	66/68	*Lactobacillus plantarum, Lactobacillus acidophilus, Bifidobacterium longum*	2.6 × 10^14^ CFU 2 g/day	Placebo	6 days prior to LR until 10 days post LR (16 days)	39/66, 60/68 FU POD 10	62.84 ± 17.17; 65.62 ± 18.18/60.16 ± 16.20	RCT
**Rayes et al., 2012 [47]**	Germany	19/33	9/10	*Pediacoccus pentosaceus,* *Leuconostoc mesenteroides Lactobacillus paracasei* subspecies *paracasei*, *Lactobacillus plantarum*; prebiotic: bioactive fibers: betaglucan, inulin, pectin, and resistant starch	probiotic: 10^10^ CFU, prebiotic: 10 g 2×/day; powder	Placebo/fibers	1 day prior to LR until 10 days post LR (11 days)	3/9, 2/10 FU POD 14	60.05 ± 13.89; 61 ± 16/59 ± 11	RCT
**Sugawara et al., 2006 [49]**	Japan	81/101	41/40	*Lactobacillus casei* strain Shirota, *Bifidobacterium breve* strain Yakult; prebiotic: galactooligosaccharides	Pre-LR: 4 × 10^10^, 80 mL 1 × 10^10^, 100 mL prebiotic: 15 g 1×/day Post-LR: 1 × 10^8^/g 1 × 10^8^/g 3 g/day prebiotic: 10 g 1×/day	No intervention, syn biotics administered only post LR	14 days prior to LR and 14 days post LR (28 days)	5/41, 12/40 FU POD 30	63.15 ± 8.84; 63.1 ± 7.9/ 63.2 ± 9.8	RCT
**Usami et al., 2011 [48]**	Japan	61/67	32/29	*Lactobacillus casei* strain Shirota, *Bifidobacterium breve* strain Yakult; prebiotic: galactooligosaccharides	1 × 10^8^/g 1 × 10^8^/g 3 g/day prebiotic: 10 g 1×/day	No intervention	14 days prior to LR and 11 days post LR (25 days)	0/32, 5/29 FU POD 30	65.42 ± 9.86; 62.1 ± 10.2/ 69.1 ± 8.0	RCT
**Rifatbegovic et al., 2010 [46]**	Bosnia	120/120	60/60	*Lactobacillus plantarum* 2362, *L. paracasei* subsp *paracasei* 19, *Pediacoccus pentoseceus* 5-33:3 and 32-77:1, *L. raffinolactis* and fibers	NR	No intervention	3 days prior to LR until 7 days post LR (10 days)	NR FU POD 14	NR	Prospective non- randomized, NOS: 7
**Iida et al., 2020 [50]**	Japan	284/324	115/169	*Clostridium butyricum* and fibers	6 g/day 12 g/day	No intervention	30 days prior to LR until 1 days prior to LR (30 days)	19/115, 40/169 FU POD 3	67.2 ± NR; 68.2 ± 11.6/ 66.2 ± 12.6	Prospective non- randomized, NOS: 8

LT, liver transplantation; LR, liver resection; LDLT, living donor liver transplantation; AST, aspartate-aminotransferase; ALT, alanine-aminotransferase; GLDH, glutamate-dehydrogenase; SSI, surgical site infection; FU, follow-up; POD, postoperative day; CFU, colony forming units; PA, patients analyzed; PE, patients enrolled.

**Table 2 nutrients-12-02461-t002:** Studies on perioperative pro-/synbiotics and their effect on liver function.

Study (Ref.)	Results Pro-Synbiotic/Control Perioperative Liver Function	Assessment of Liver Function	Study Design
**Liver Transplantation (LT)**			
Rayes et al., 2002 [40]	No difference AST, ALT, GGT, AP	FU POD 13	RCT
Rayes et al., 2005 [41]	No difference AST, ALT, GGT, AP	FU POD 30	RCT
Grat et al., 2017 [39]	-ALT (IU/l) 398.8 ± 307.68/441.05 ± 432.72 -AST (IU/l) 140.4 ± 123.59/105.6 ± 62.92 -Bili (mg/dl) 2.5 ± 1.91/2.9 ± 2.59 -INR 1.05 ± 0.10/1.16 ± 0.18	FU POD 5	RCT
**Liver Resection (LR)**			
Liu et al., 2015 [45]	-ALT (U/l) 32.62 ± 18.86/35.68 ± 15.26 -AST (U/l) 28.22 ± 18.86/29.68 ± 16.56 -ALT (U/l) 36.28 ± 18.92/56.20 ± 18.16 -AST (U/l) 36.18 ± 21.52/45.62 ± 22.68	Prior to LR FU POD 10	RCT
Rayes et al., 2012 [47]	^13^C methacetin test (LiMAx (%)) 160 ± 45/135 ± 60 ^13^C methacetin test (LiMAx (%)) 260 ± 85/240 ± 80 -AST (U/l) 110/90 -ALT (U/l) 210/150 -GLDH (U/l) 70/35 -AST (U/l) 80/60 -ALT (U/l) 100/95 -GLDH (U/l) 30/25 -AST (U/l) 65/50 -ALT (U/l) 100/70 -GLDH (U/l) 2	FU POD 5 FU POD 14 FU POD 5 FU POD 10 FU POD 14	RCT
Usami et al., 2011 [48]	No difference AST, ALT, bilirubin	FU POD 15	RCT
Rifatbegovic et al., 2010 [46]	-ALT (U/l) 50 ± 5/68 ± 7 -Bili (mcmol/l) 17 ± 0.8/31.3 ± 1.5 -Indocyaningreen test 1.15/1.425	FU POD 14	Prospective non-randomized, NOS: 7
Iida et al., 2020 [50]	PostLR liver failure (grade) -A: 8/115, 13/169 -B: 1/115, 16/169 -C: 1/115, 4/169 -ALT (IU/l) prior to synbiotics–after synbiotics (*n* = 115): 23 (16, 31)–20 (13, 34) -AST (IU/l) prior to synbiotics–after synbiotics (*n* = 115): 27 (22, 43)–26 (20, 34) -Bilirubin (mg/dl) prior to synbiotics–after synbiotics (*n* = 115): 0.7 (0.5, 0.8)–0.6 (0.4, 0.8) Prothrombin activity (%) prior to synbiotics–after synbiotics (*n* = 115): 96 (87, 109)–96 (85, 103)	FU POD 3	Prospective non-randomized, NOS: 8

Data are expressed as mean ± standard deviation (SD) [39,45,46,47], median (interquartile range) [50]; LT, liver transplantation; LR, liver resection; LDLT, living donor liver transplantation; AST, aspartate-aminotransferase; ALT, alanine-aminotransferase; GLDH, glutamate-dehydrogenase; SSI, surgical site infection; FU, follow-up; POD, postoperative day; LiMAx, maximum liver function capacity; Characteristics of studies as defined in Table 1.

**Table 3 nutrients-12-02461-t003:** Studies on perioperative pro-/synbiotics and their effect on changes of the gut microbiota.

Study (Ref.)	Gut Microbiota Changes
**Liver Transplantation (LT)**	
Eguchi et al., 2011 [42]	No significant changes between the groups *Enterococcus* sp. evident in both groups in 25% of the immunosuppressed patients
Grat et al., 2017 [39]	Probiotic group: *Bacteroides* sp. (*p* = 0.008), *Enterococcus* sp. (*p* = 0.04) significantly increased in comparison to pre-trial values as compared to control group
**Liver Resection (LR)**	
Kanazawa et al., 2005 [44]	Synbiotic group: *Lactobacillus* and *Bifidobacterium* increased postoperatively in comparison to controls (*p* < 0.05) Control group: *Enterobacteriaceae*, *Pseudomonas*, Candida increased in comparison to synbiotic group (*p* < 0.05) Enterococci increased postoperatively in both groups
Sugawara et al., 2006 [49]	Pre-and post-operative probiotic group: *Bifidobacterium* significantly increased after preoperative treatment (*p* < 0.05) Anaerobic bacteria numbers were unchanged before and after surgery between the two groups
Usami et al., 2011 [48]	*Bacteroidaceae*, *Bifidobacterium* decreased, Candida increased 1 week postoperatively, resembled preoperative values after 2 weeks No differences concerning liver function

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
