# Peer review of "Effects of both Pro- and Synbiotics in Liver Surgery and Transplantation with Special Focus on the Gut–Liver Axis—A Systematic Review and Meta-Analysis"

_nutrients, 2020, doi:10.3390/nu12082461_

Round 1

Reviewer 1 Report

The manuscript by Kahn and colleagues is a systematic review and meta-analysis of the current literature regarding the effects of pro-/synbiotic in liver surgery. The manuscript is well written, and in my opinion is really interesting, also because it covers an important aspect of the current surgical therapy still debated.

Reviewer 2 Report

Dear Authors

The Editors of Nutrients have invited me to review your manuscript, which I have found to be of excellent quality in terms of its contribution to filling the knowledge gap in this subject, its angle, and its methodological quality. The presentation is also impressive. 

I have provided some specific comments with regards to the claim that "probiotics also have an impact on neuromodulation by expressing cannabinoid and µ-opioid receptors on the epithelial cells." which I hope you find useful. 

All the best with the rest of the publication process.

With kindest regards,

The MDPI Reviewer

Reviewer 3 Report

The manuscript is in line with the currently popular trend in medicine to look how the bacterial supplments affect multiple entities. Tha paper is really good with data basicly supporting the conclusions from existing literature, but some important issue exists that I want to point

Results:

  1. Provide the dosage and form of probiotics (capsules, powder)
  2. The names of probiotics should be written in italics
  3. The study population mentioned in Table 1 refers to persons randomized/analyzed?
  4. Sections 3.4 and 3.5 are difficult to follow. Please condider to present data in a table or rephrase these paragraphs to leave only data that are not presented in forest plots – the images give enough information of infection rates
  5. Please conduct metaanalysis for biochemical parameters of liver function
  6. If possible conduct meteregression with respect to age of participants and duration of supplementation
  7. Is there any data regarding microbiome analyses?

Discussion

Please provide a paragraph on strain specificity regarding your study
